# Inhibitory Effects of Breast Milk-Derived *Lactobacillus rhamnosus* Probio-M9 on Colitis-Associated Carcinogenesis by Restoration of the Gut Microbiota in a Mouse Model

**DOI:** 10.3390/nu13041143

**Published:** 2021-03-30

**Authors:** Haiyan Xu, Keizo Hiraishi, Lin-Hai Kurahara, Yuko Nakano-Narusawa, Xiaodong Li, Yaopeng Hu, Yoko Matsuda, Heping Zhang, Katsuya Hirano

**Affiliations:** 1Key Laboratory of Dairy Biotechnology and Engineering, Ministry of Education, Inner Mongolia Agricultural University, Hohhot 010018, China; weiliangxhy@163.com (H.X.); hepingdd@vip.sina.com (H.Z.); 2Department of Cardiovascular Physiology, Faculty of Medicine, Kagawa University, Kagawa 761-0793, Japan; kzohiraishi@med.kagawa-u.ac.jp (K.H.); lixiaodong@med.kagawa-u.ac.jp (X.L.); khirano@med.kagawa-u.ac.jp (K.H.); 3Oncology Pathology, Department of Pathology and Host-Defence, Faculty of Medicine, Kagawa University, Kagawa 761-0793, Japan; y_nakano@med.kagawa-u.ac.jp (Y.N.-N.); mazdayoko@gmail.com (Y.M.); 4Department of Physiology, Faculty of Medicine, Fukuoka University, Fukuoka 814-0180, Japan; huyaopeng@fukuoka-u.ac.jp

**Keywords:** inflammatory bowel disease, colitis-associated tumorigenesis, probiotic, *Lactobacillus rhamnosus* M9

## Abstract

Chronic inflammation is a risk factor for colorectal cancer, and inflammatory cytokines secreted from inflammatory cells and active oxygen facilitate tumorigenesis. Intestinal bacteria are thought to regulate tumorigenesis. The longer the breastfeeding period, the lower is the risk of inflammatory bowel disease. Here, we investigated preventive effects of the probiotic *Lactobacillus rhamnosus* M9 (Probio-M9) on colitis-associated tumorigenesis. An inflammatory colorectal tumor model was established using a 6-week-old male C57BL/6NCrSlc mouse, which was intraperitoneally administered with azoxymethane (AOM: 12 mg/kg body weight). On weeks 2 and 4, 2% dextran sulfate sodium (DSS) was administered to mice for 7 days through drinking water. On weeks 8 and 10, Probio-M9 (2 × 10^9^/day) was orally administered for 7 days. Animals were sacrificed at 20 weeks after AOM administration and immunohistochemical staining and Western blotting was performed. The α-diversity of microflora (Shannon index), principal coordinate analysis, and distribution of intestinal bacterium genera and metabolic pathways were compared. The AOM/DSS group showed weight loss, diarrhea, intestinal shortening, increased number of colon tumors, proliferating tumorigenesis, increased inflammation score, fibrosis, increased CD68+, or CD163+ macrophage cells in the subserosal layer of non-tumor areas. Inflammation and tumorigenesis ameliorated after Probio-M9 treatment. Fecal microbial functions were altered by AOM/DSS treatment. Probio-M9 significantly upregulated the fecal microbial diversity and reversed fecal microbial functions. Thus, Probio-M9 could suppress tumor formation in the large intestine by regulating the intestinal environment and ameliorating inflammation, suggesting its therapeutic potential for treatment of inflammation and colitis-associated tumorigenesis.

## 1. Introduction

Colorectal cancer can be divided into two types, namely, colitis-associated cancer (CAC) and sporadic colorectal cancer [1]. Prolonged inflammatory bowel disease (IBD) is one of the major risks related to CAC development, with an incidence of 2% after a 10-year history of IBD that may rise up to 18% after 30 years [2]. Here, we examined the effects of probiotics on carcinogenic progression by intervening in the long process from IBD to CAC [3].

The environment of the intestinal microbiota is known to significantly influence the pathogenesis of IBD and CAC in both animal models and humans [4,5,6]. Dysregulation in the bacterial phyla involved in the production of short-chain fatty acids was observed in patients with IBD. Many studies have shown that oral administration of a commercial probiotic cocktail containing probiotic VSL#3 (four strains of lactobacilli, three strains of *Bifidobacterium*, and one strain of *Streptococcus*) could reduce chronic inflammation and delay the onset of carcinoma in a mouse model of CAC [7]. Therefore, administration of probiotics to restore the balance between the microbiome community is thought to be effective in patients with IBD and CAC [8,9]. These studies support the view that probiotics prevent inflammation and carcinogenesis. However, the mechanisms underlying the therapeutic effects of probiotic treatment in CAC are not clearly elucidated.

Breast milk is the main source of nutrition of the growing baby. The longer the breastfeeding period, the lower is the risk of gastrointestinal inflammation. Breastfeeding in infancy protects against the development of Crohn’s disease and ulcerative colitis [10]. This association has been observed in all ethnic groups, and the magnitude of protection in Crohn’s disease was reported to be significantly higher among Asians than Caucasians. Breastfeeding duration showed a dose-dependent association, with the strongest decrease in the risk of Crohn’s disease and ulcerative colitis in those breastfed for at least 12 months as compared to those breastfed for 3 or 6 months [10]. Although the composition and function of gut microbiota is easily disrupted throughout life, the microbiome is unstable and sensitive to external environmental exposure in early infancy. Factors that contribute to the microbiome in early childhood can have a strong impact on the development of diseases such as IBD. The Inner Mongolia University of Agriculture isolated and identified lactic acid bacteria from the human colostrum and screened potential probiotic strains. *Lactobacillus rhamnosus* Probio-M9 [11] can potentially function as a probiotic and improve the intestinal environment by increasing the diversity of the gut flora and regulating metabolic pathways. The effects of Probio-M9 on colorectal tumors have not been assessed yet. It is widely known that changes in the microflora community may act as driving forces during the stepwise progression from inflammation to dysplasia to adenocarcinoma. Therefore, the possibility of suppressing tumor progression by manipulating intestinal flora is logical.

In IBD, the interleukin (IL)-6/signal transducer and activator of transcription 3 (STAT3) signaling is an important regulator of proliferation of tumor cells [12]. Phosphoinositide 3-kinases (PI3Ks) and the downstream serine/threonine kinase protein kinase B (Akt) regulate inflammatory responses, cell cycle entry, survival, and apoptosis. During cell proliferation, Ki-67 and proliferating cell nuclear antigen (PCNA) are overexpressed as cell proliferation marker proteins in the cancer tissue [13]. Polarized macrophages are mainly divided into two types, classically activated macrophages (M1) and alternately activated macrophages (M2). M1 macrophages are characterized by the secretion of pro-inflammatory cytokines IL-1β, IL-6, IL-12 and tumor necrosis factor (TNF)-α, while M2 macrophages induce a weaker immune response and reduce related proinflammatory fibrosis. However, M1/M2 polarization and functions in IBD carcinogenesis are still unclear. Tumor-associated macrophage-specific markers CD68 and CD163 were used as a marker of M1 and M2, respectively in this study. CD68+ macrophages are used as an indicator of inflammation progression, while CD163+ macrophages are associated with early local recurrence and reduced survival times [14].

A tumor is a disease related to glucose, lipid, and protein or amino acid metabolism disorders. The host microbial metabolism is important for tumor moderation. It has been reported that the microbiota induces low level of immune system activation that promotes inflammation in the tumor microenvironment [15]. High-protein diets, for example, can reduce the beneficial microbial species and metabolites and downregulate the genes involved in immunoprotection [16]. Few trials have confirmed that several bacteria, including *Bifidobacterium* and *Lactobacillus* spp., exhibit anticancer properties in preclinical studies through different mechanisms such as by modulating the host immunity [17], inactivating carcinogenic toxins [18] and producing anticarcinogenic compounds [19].

The purpose of this study was to evaluate the protective effect of Probio-M9 against azoxymethane (AOM)/dextran sulfate sodium (DSS)-induced CAC in a mouse model. We identified several significantly altered intestinal bacterial communities that were involved in mediating protective effects of Probio-M9 treatment. Our results confirm that Probio-M9 has the ability to suppress colon carcinogenesis in mice and provide evidence of its clinical application to reduce the prevalence of CAC in patients with recurrent IBD.

## 2. Materials and Methods

### 2.1. Materials

DSS (M.W. 36,000 to 50,000; CAS Number: 9011-18-1; MP Biomedicals, Inc., Santa Ana, CA, USA) and AOM (A5486 from Sigma-Aldrich Co., St. Louis, MO, USA) were used to establish the CAC model mice. *L. rhamnosus* Probio-M9 was provided by the Key Laboratory of Dairy Biotechnology and Engineering, Ministry of Education, Inner Mongolia Agricultural University, P. R. China. PCNA antibodies were procured from Dako; Ki67 (SP6), CD68 (polyclonal), CD163 (EPR19518) and CD3 (SP7) were supplied by Abcam (Cambridge, UK).

### 2.2. Animal Experiments

Eight-week-old male C57BL/6NCrSlc mice were intraperitoneally administered with AOM (12 mg/kg body weight (BW), Figure 1A). On weeks 2 and 4, mice were exposed to 2% DSS through drinking water for 7 days. On weeks 8 and 10, Probio-M9 (2 × 10^9^/day) was orally administered to each mouse for 7 days (two cycles of M9 administration for a total of 14 days). Body weight and stool consistency were checked once per week. Stool consistency was scored as follows: 0, normal; 1, slightly loose stool; 2, severely loose stool; 3, diarrhea. Animals were sacrificed under anesthesia 20 weeks after AOM administration. The experimental design was approved by the Animal Care and Use Committee of Kagawa University, Japan (approval number 19652) and Fukuoka University, Japan (approval number 1709099).

### 2.3. Histopathological Evaluation

The colon was excised, and the open long axis was cut. The colon length and number of colon tumors were macroscopically analyzed. The tissues were then fixed in formalin overnight, routinely processed for paraffin embedding, and cut into 3-μm thick sections for hematoxylin-eosin (HE) and Masson’s trichrome (MT) staining.

Histology score was evaluated by a pathologist (Y.M.) under 200× magnification after HE staining. The inflammation score was calculated as the sum of I, E, R, C and P. Criteria are summarized in Table 1 [20].

Fibrosis score was obtained by quantifying the range of the blue area of MT staining in a blinded manner. Fibrosis score was categorized as 0%, 1–25%, 26–50%, 51–75%, or 76–100% of the submucosa affected that corresponded to a fibrosis score of 0, 1, 2, 3, and 4, respectively.

### 2.4. Immunohistochemistry

Mouse tissue specimens were immunostained by the labeled streptavidin-biotin method using the Ventana Discovery staining system (Ventana Medical Systems, Oro Valley, AZ, USA) with DISCOVERY DAB Map Detection Kit (Roche, Basel, Switzerland). The anti-mouse CD3 monoclonal antibody (1:150 dilution); the anti-mouse CD68 polyclonal antibody (1:100 dilution), anti-mouse CD163 monoclonal antibody (1:500 dilution), and anti-mouse Ki-67 and PCNA monoclonal antibodies (1:200 dilution) used were diluted with DISCOVERY Antibody Diluent (Roche, Switzerland) [21]. For CD163 samples, cell conditioning 1 (CC1, pH 8.5, Roche) antigen activation was performed for 60 min at 100 °C. For other samples, RiboCC (pH 6.0, Roche, Switzerland) antigen activation was performed for 60 min at 100 °C. The reaction times of the primary antibody were CD3, 60 min and the others, 12 h. The reaction time of the second antibody were CD3, 60 min and the others, 32 min. Counterstaining was performed with hematoxylin (Roche, Switzerland). Negative control specimens were stained without primary antibodies. Images of non-tumor subserosa were captured by Olympus BX-51/DP-72 under 400× magnification, and the number of CD68- and CD163+ cells per high power field (HPF, 0.196 mm^2^) was counted.

### 2.5. Real-Time Polymerase Chain Reaction (PCR) Analysis

Colon tissue were cut and grind by homogenizer in FARB buffer from Tissue Total RNA Mini Kit (FAVORGEN, Taipei City, Taiwan), RNA purified with the Tissue Total RNA Mini Kit (FAVORGEN, Taiwan). After RNA concentration measurement, reverse transcription process was conducted with Prime Script RT Master Mix (Takara Bio Inc., Shiga, Japan). TaqMan Fast advanced Master Mix (Applied Biosystems, Foster City, CA, USA) was used for a quantitative polymerase chain reaction (PCR) process in a MicroAmp Fast 96 well Reaction Plate (Applied Biosystems, USA). The TaqMan probes, (Applied Biosystems, USA) used in this study, were interleukin-6 (IL-6; Mm00446190_m1), interleukin-17 A (IL17a; Mm00439618_m1), interferon gamma (IFN-γ; Mm01168134_m1), transforming growth factor beta 1 (TGF-β1; Mm01178820_m1), and tumor necrosis growth factor alpha (TNF-α; Mm00443258_m1). Mouse Glyceraldehyde-3-phosphate dehydrogenase (GAPDH) (Applied Biosystems, USA) was used as endogenous control. Real-time PCR reaction was performed by the Viia7 Real-Time PCR (RT-PCR) system (Applied Biosystems, USA). The protocol was: initial denaturation at 95 °C for 20 s, 70 cycles of denaturation (95 °C, 5 s) and annealing/extension (60 °C, 30 s).

### 2.6. Metagenomic Sequencing and Quality Control

The metagenomic DNA libraries were constructed with 2 μg genomic DNA according to the Illumina TruSeq DNA Sample Prep v2 Guide, and we assessed the quality of all libraries using an Agilent bioanalyzer with a DNA LabChip 1000 kit. Sequencing was performed using an Illumina NovaSeq 6000 instrument. Then, the quality control obtained raw paired-end reads were processed for based on the following criteria: (1) reads with adaptors were removed by the software Seqprep (https://github.com/jstjohn/SeqPrep, accessed on 21 January 2019); (2) we trimmed reads from their 3′ ends at a quality threshold of 30; (3) we removed reads containing more than 50% bases of low quality (Q30); (4) we removed reads shorter than 70 bp using the software Sickle (https://github.com/najoshi/sickle, accessed on 10 February 2019); and (5) we removed any reads that aligned to the mouse genome. High-quality reads were retained and used for the next analysis.

### 2.7. Taxonomy and Functional Annotation

We normalized the high-quality metagenomic shotgun sequences through the random sampling method with an in-house script using the lowest sequencing depth of all samples. We used kraken2 and Bracken for species abundance profiling with default parameters following the instructions of the software developer. We used Humann2 to perform the functional annotation.

### 2.8. Statistical Analyses

Differences in numerical variables among groups were evaluated using analysis of variance (ANOVA), and the Tukey–Kramer test was used for multiple comparisons for all pairs. A value of *p* < 0.05 was considered statistically significant.

Statistical analyses of the microbiome were mainly completed using R packages (http://www.r-project.org/, accessed on 15 April 2019) and Python. We evaluated differences in the relative abundances of taxonomic groups between samples at the gene level using the Mann–Whitney test and obtained false discovery rate (FDR) values using the Benjamini–Yekutieli method to control for multiple testing.

### 2.9. Datasets Generated by the Whole Genome Metagenomics

The sequencing result in this study was deposited in the NCBI Short Read Archive (SRA) database (https://www.ncbi.nlm.nih.gov/home/submit/) under accession numbers PRJNA694773.

## 3. Results

### 3.1. Probio-M9 Administration Suppresses Inflammatory Carcinogenesis

The AOM/DSS group showed a significant decrease in the BW as compared with the vehicle group, while the AOM/DSS+M9 group showed a tendency of improvement in BW loss but without any significant difference (Figure 1A, *p* < 0.01 vs. vehicle, after 15 and 20 weeks of AOM administration). The AOM/DSS group showed severe stool consistency (Figure 1B, *p* < 0.01 vs. vehicle group, 20 weeks), which was ameliorated after Probio-M9 treatment (Figure 1B, *p* < 0.05 vs. AOM/DSS group). The AOM/DSS group had an increase in the number of colon tumors (Figure 1C, *p* < 0.01 vs. vehicle group), which were suppressed in the AOM/DSS+M9 group (*p* < 0.05 vs. AOM/DSS group). The estimated tumor size was significantly lower in the AOM/DSS+M9 group than in the AOM/DSS group (Figure 1C, *p* < 0.05). The detail of tumor number and size are summarized in Appendix A.

In the AOM/DSS model mice, precancerous lesions were observed 10 weeks after AOM administration, and tumorigenesis with intestinal shortening was obvious at 20 weeks after AOM treatment (Figure 2A,B). We examined the proliferation activity of PCNA+ cells, which is associated with tumorigenesis, and found it to be significantly higher in the tumor areas of AOM/DSS mice. These changes were suppressed by Probio-M9 administration (Figure 2B,C). In the vehicle mice, β-catenin was localized in the cell membrane of epithelial cells, whereas in the tumor area, nuclear translocation was observed overall in AOM/DSS and AOM/DSS+M9 mice (Figure 2C). Western blot analysis of tumor tissues was performed, and the expression of p-STAT3 and p-Akt was found to be significantly suppressed after Probio-M9 administration (Figure 2D).

### 3.2. Analysis of the Therapeutic Effect of Probio-M9 by Pathological Immunostaining in Non-Tumor Areas

To evaluate the anti-inflammatory effects of Probio-M9, the inflammatory score of non-tumor areas was analyzed by evaluating inflammation extent, regeneration, crypt damage, and percent involvement by H&E staining. AOM/DSS treatment significantly induced inflammatory damage in the intestine, and this effect was ameliorated by Probio-M9 treatment (Figure 3A). MT staining showed that AOM/DSS administration induced fibrosis mainly in the submucosal layer and partially in the mucosal layers (Figure 3B, arrows), which was significantly ameliorated by Probio-M9 treatment (*p* < 0.05 vs. AOM/DSS group). The vehicle group showed fewer PCNA+ cells in the mucosal areas of non-tumor regions. PCNA+ cell number significantly increased in the AOM/DSS group and was much higher in the AOM/DSS+M9 group (Figure 3C). We performed RT-PCR of the cytokines IL-6, IL-17a, TNF-α, and INF-γ in distal colon. We were able to obtain upregulation in some AOM/DSS mice, but we could not confirm statistically significant differences (Appendix A) [22,23].

We examined another proliferative marker, Ki-67, in these non-tumor areas, and found that the vehicle group had fewer Ki67+ cells in the mucosal area. The number of these cells significantly increased in the AOM/DSS+M9 group compared with that in the AOM/DSS group (Figure 4, arrows).

To analyze the localization and contribution of macrophages, we performed CD68 and CD163 immunostaining of the colon tissue. We also evaluated CD68+ and CD163+ cells in the mucosal layer but found no statistically significant changes among the three groups (data not shown). However, the AOM/DSS group had several CD68+ cells in the subserosal layer of the non-tumor regions that were suppressed by Probio-M9 administration (Figure 4, arrows). CD163+ cells are located in the normal mucosal, submucosal, muscle, and subserosa areas. The AOM/DSS group showed an increase in the population of CD163+ cells in the subserosa layer of non-tumor areas, and this effect was recovered by Probio-M9 administration (Figure 4, arrows).

To analyze the contribution of T and B cells in this CAC model, we performed CD3 and CD20 immunostaining. The number of CD3+ cells (T cells) are only slightly expressed in non-tumor tissues, and expression in the lymphoid follicle and tumor area was not significantly different between the AOM/DSS and AOM/DSS+M9 groups (Appendix A). The number of CD20+ (B cell) cells located in the lymphoid follicle was not significantly different between the three groups (data not shown).

### 3.3. Changes in the Fecal Microbiota after Treatment

Although a slight difference (*p* > 0.05) was found in the Shannon index between the vehicle and AOM/DSS groups at the end of the experiment, the Shannon index was significantly higher in Probio-M9 group than in the AOM/DSS group (*p* < 0.05). This observation suggests that the intake of probiotics with AOM/DSS could help lower the effect of AOM/DSS in some animals (Figure 5A). These results might suggest the beneficial effect of probiotics on the gut microbial diversity.

To delineate the effect of probiotic treatment on the structure of the fecal microbiota, a principal coordinate analysis (PCoA) was performed on the basis of the weighted UniFrac distances at the end of the experiment (Figure 5B). No obvious clustering pattern was observed in the PCoA score plot between the three groups, but one of the three samples from the AOM/DSS group was obviously far away from the cluster pattern on the PCoA score plots (Figure 5B). The microbial structure showed a higher divergence between AOM/DSS and the other two groups, suggesting that AOM/DSS application induced obvious changes in the gut microbiota.

The heatmap showed differences in the microbial composition between the three groups at the genus level (Figure 5C). The Probio-M9 and vehicle groups clustered more closely, as a differential abundant genus was observed between the AOM/DSS group and the other groups. The AOM/DSS group comprised significantly lower *Blautia*, *Akkermansia*, *Lactobacillus*, *Clostridium*, *Bifidobacterium*, and *Eubacterium* (all *p* < 0.05) and significantly higher *Muribaculum*, *Bacteroides*, *Alistipes*, and *Butyricimonas* than the other two groups, suggesting that AOM/DSS treatment induced changes at the genus level in the gut microbiota community. However, Probio-M9 together with AOM/DSS could increase the abundance of these genera, especially *Blautia*, *Akkermansia*, and *Bifidobacterium*, indicating that the probiotic could balance the intestinal microbiota in an AOM/DSS mouse model.

### 3.4. Changes in the Fecal Microbial Function after Treatment

We investigated the functional changes in the gut microbiota through analyzing the expression of functional genes by Humann2. The differences in the functional pathways were compared between the three groups (Figure 6). Some pathways such as the superpathway of glycol metabolism and degradation, biotin biosynthesis II, allantoin degradation to glyoxylate III, and rubisco shunt were depleted in the AOM/DSS group. AOM/DSS group had significant upregulation in phytate degradation I, starch biosynthesis, CMP-N-acetylneuraminate biosynthesis I (CMP: Cytosine nucleotides), nitrate reduction VI (assimilatory), sucrose degradation IV (sucrose phosphorylase), and L-lysine fermentation to acetate and butanoate as compared to the other groups (*p* < 0.05) and significant downregulation in pathways such as superpathway of phylloquinol biosynthesis, 1,4-dihydroxy-2-naphthoate biosynthesis I, and superpathway of menaquinol-11 biosynthesis (*p* < 0.05). These results indicate that the functional pathways in Probio-M9 group were more similar to those in the vehicle group and that the gut microbiota of the AOM/DSS group showed differential functions as compared to those of the other two groups.

## 4. Discussion

This study confirmed that the administration of Probio-M9 reduced inflammation, decreased tumor number and average size, and ameliorated pro-inflammatory macrophages and fibrosis in an AOM/DSS mouse model. These results are consistent with many previous reports focusing on probiotics. In particular, we used fewer probiotics as compared to the study with VSL#3 administration (VSL#3 vs. Probio-M9: 12 weeks vs. 2 weeks) and achieved significant inhibition of inflammation and tumor formation in our AOM/DSS mouse model [7].

PCNA and Ki67 immunostaining experiments confirmed that Probio-M9 administration suppressed the abnormal growth signal in tumor areas and accelerated mucosal repair in non-tumor areas. Improvement in the intestinal environment by Probio-M9 mediated a protective effect against inflammation and regenerated the mucosa. The expression of macrophages M1(CD68+)/M2(CD163+) in the serosa was suppressed after the administration of Probio-M9, indicating improvement of inflammation. Along with the inhibition of these inflammations, the phosphorylation of Akt and STAT3, which indicate carcinogenic signals, were also suppressed.

The suppression of these inflammatory carcinogenesis is considered to be due to the improvement of the intestinal environment. Our data show that AOM/DSS significantly lowered the gut microbiota diversity and that Probio-M9 did not induce any obvious fluctuation in the Shannon diversity index. An interesting effect of probiotic treatment in combination with AOM/DSS was the stabilization of the intestinal microbiota at the end of the experiment. In consistent with the changes in the α-diversity, the weighted UniFrac distances of the three groups did not significantly differ and the distance between AOM/DSS group and the other groups was higher on the PCoA plot. The greater similarity in the intestinal microbial diversity between Probio-M9 and vehicle groups suggests probiotic-dependent modulation. The α-diversity defined by richness and evenness is an ecological description of an individual community. The α-diversity of the gut microbiota in human IBD patients is typically reduced [24,25,26].

Metagenomic sequencing showed that AOM/DSS decreased the abundance of *Blautia*, *Akkermansia*, *Clostridium*, *Bifidobacterium*, and *Eubacterium* and facilitated the growth of *Muribaculum*, *Bacteroides,* and *Alistipes*. However, Probio-M9 seemed to counteract the role of AOM/DSS by increasing the population of *Akkermansia*, *Bifidobacterium*, and *Lactobacillus*. The fecal microbiota composition of the Probio-M9 group was clustered more closely with that of the vehicle group. Herein, *Akkermansia* comprised only *Akkermansia muciniphila*, which is an intestinal symbiont colonizing the mucosal layer and is considered a promising probiotic candidate known to have a vital value in improving the host metabolic functions and immune responses [27]. The *Blautia* species (*Clostridium* cluster XIVa) is also well known as a part of the butyrate-producing bacteria of the gut microbiota. Butyrate is a bacterial metabolite that could account for the beneficial roles of these bacteria in glucose metabolism and obesity-associated inflammation [28,29]. Furthermore, the *Bifidobacterium* and *Lactobacillus* population increased in fecal samples after administration of Probio-M9, suggesting that Probio-M9 administration following the onset of AOM/DSS-induced colitis could promote a healthy intestinal bacterial community. The genera *Bifidobacterium* and *Lactobacillus* can have multiple effects on the host by conferring competitive exclusion of pathogens, normalizing altered microbiota, regulating intestinal transit time and short-chain fatty acid production, reinforcing mucosal barrier function, and interacting with antigen-presenting dendritic cells [30]. Interestingly, Probio-M9 is a strain of *L. rhamnosus*, and the increase in the abundance of *Lactobacillus* in the fecal intestinal microbiota at the end of the experiment suggests that this may be due to the administration of probio-M9. *Muribaculum* population was higher in AOM/DSS group than in probiotic and vehicle groups, and is reported to be enriched in mice with T cell-induced colitis [31]. A study revealed that *Bacteroides* were enriched preceding overt inflammation [32]. Overall, AOM/DSS treatment induced a decrease in some beneficial bacteria. The positive effect of probiotics on the intestinal microbiota of the host was mediated through the regulation of the proportion of beneficial and harmful bacteria by an increase in beneficial microorganisms and a decrease in harmful bacteria.

Our study found that some microbial functions in the AOM/DSS group were different from those in the vehicle group, such as biotin biosynthesis II, sucrose degradation IV, and starch biosynthesis. These functions in the probiotic-treated group were similar to those in the vehicle group. Therefore, Probio-M9 regulated the gut microbiota composition and influenced their metabolism, especially sugar metabolism. Sugar metabolism may have a more profound role in driving tumors, as reported in a *Drosophila* genetic model [33].

## Figures and Tables

**Figure 1 nutrients-13-01143-f001:**
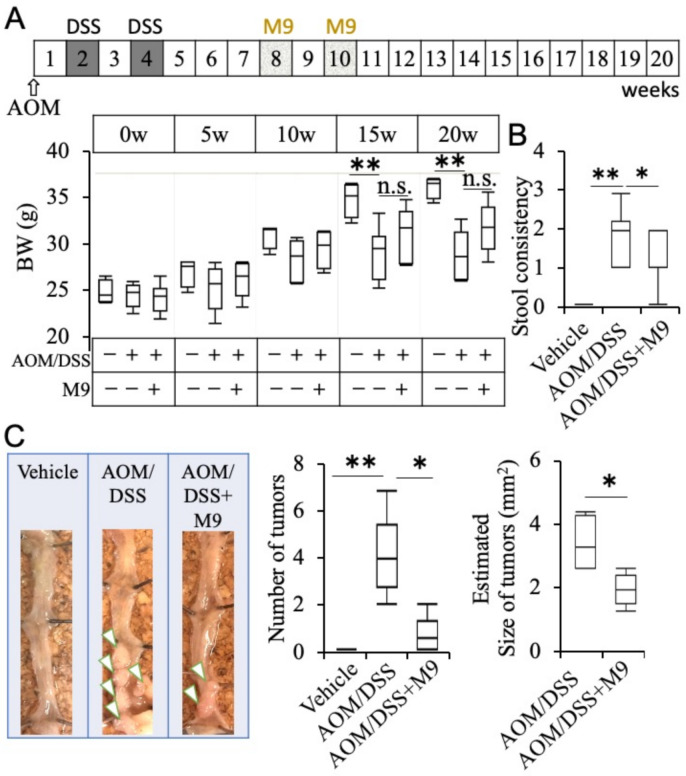
Treatment of Probio-M9 ameliorated azoxymethane/dextran sulfate sodium (AOM/DSS)-induced inflammation in a colitis-associated cancer (CAC) mouse model. (**A**) Treatment schedule and body weight (BW) changes for each group over the period of the experiment. Data are indicated in box-plot graphs. (**B**) Stool consistency score was interpreted as follows: 0, normal; 1, slightly loose stool; 2, severely loose stool; 3, diarrhea. (**C**) Representative images of colon morphology, total number of tumor, and estimated tumor size per mouse. *n* = 10. * *p* < 0.05; ** *p* < 0.01. n.s. means not significantly error.

**Figure 2 nutrients-13-01143-f002:**
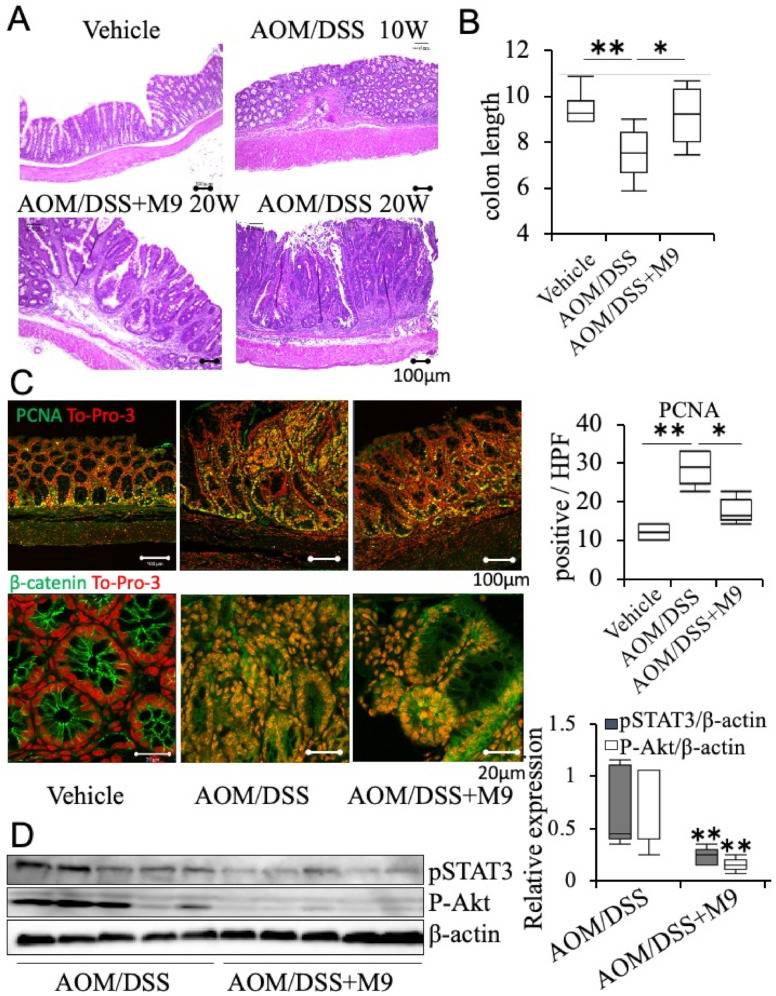
Treatment with Probio-M9 ameliorated AOM/DSS-induced tumorigenesis. (**A**) Representative images of colon morphology, total number of tumor, and estimated tumor size per mouse. (**B**) Statistics of colon length. *n* = 10. (**C**) Representative image of immunohistochemistry staining for proliferating cell nuclear antigen (PCNA) in tumor areas. Positive cell number in the mucosa per high power field (HPF: 0.196 mm^2^) was counted and summarized in graphs. *n* = 5. Representative image of immunohistochemistry staining for β-catenin in vehicle intestine and tumor areas of AOM/DSS, and AOM/DSS+M9. To-Pro-3 was used to label the nucleus. (**D**) Representative image of immunoblotting experiments for p-STAT3 and p-Akt. β-Actin was used as an internal control. Results are summarized in graphs. *n* = 5. * *p* < 0.05; ** *p* < 0.01. scale bar: 100 μm.

**Figure 3 nutrients-13-01143-f003:**
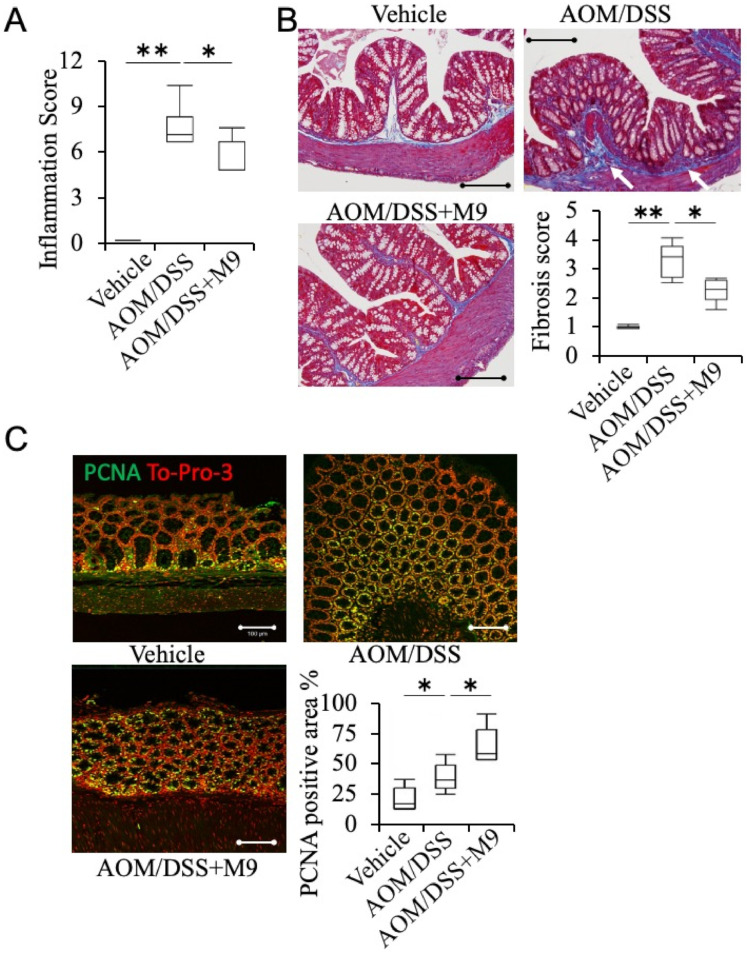
Treatment of Probio-M9 ameliorated AOM/DSS-induced fibrosis and accelerated tissue repair in non-tumor areas. (**A**) Inflammation score in non-tumor areas was evaluated from hematoxylin-eosin (HE) histological sections of the colons. *n* = 10. (**B**) Representative images of the non-tumor areas of the colon evaluated with Masson’s trichrome (MT) staining and the corresponding fibrosis scores. Averaged fibrosis score was grouped as 0%, 1–25%, 26–50%, 51–75%, or 76–100% of the submucosa affected corresponding to a fibrosis score of 0, 1, 2, 3, and 4, respectively. *n* = 10. Scale bar: 200 μm. (**C**) Representative image of immunohistochemistry staining of PCNA in non-tumor areas. To-Pro-3 was used to label the nucleus. The ratio of the expression area in the mucosal layer is shown in the graph. *n* = 5. * *p* < 0.05; ** *p* < 0.01. Scale bar: 100 μm.

**Figure 4 nutrients-13-01143-f004:**
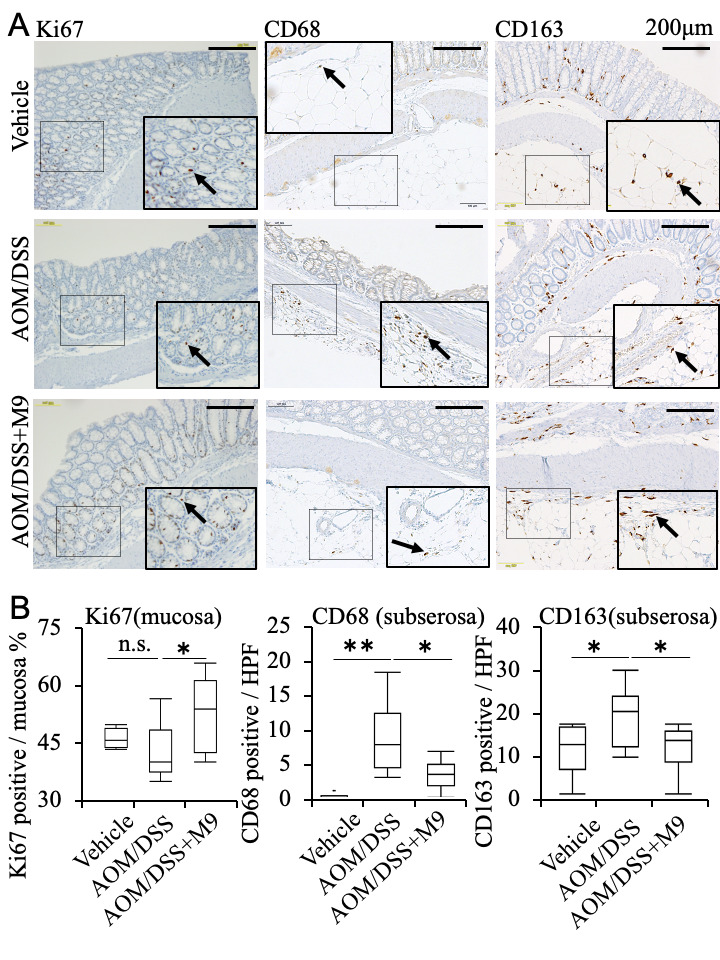
The expression of selected molecules in an AOM/DSS-induced mouse model of colitis-associated cancer. (**A**) Representative image of immunohistochemistry staining for Ki67, CD68, and CD163 in the non-tumor areas (arrows). Positive signals for Ki67 in cell nuclei were detected, and both CD68 and CD163 markers were observed in the cell cytoplasm. (**B**) The percentage of Ki67+ staining in the mucosal cell nucleus was scored. The number of cells positive in the subserosa per high power field (HPF: 0.196 mm^2^) was counted and summarized in graphs. *n* = 10. * *p* < 0.05; ** *p* < 0.01. Scale bar: 200 μm. n.s. means not significantly error.

**Figure 5 nutrients-13-01143-f005:**
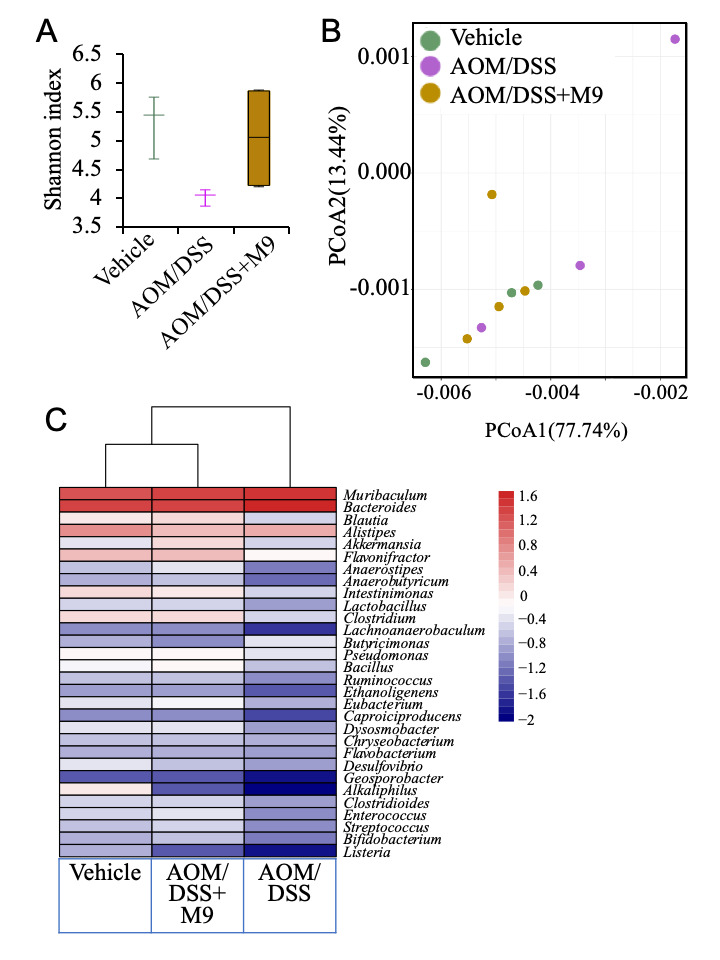
The difference in the fecal microbiota at week 17. (**A**) The difference in the Shannon index of three groups at week 17. (**B**) Principal coordinates analysis (PCoA) of the gut microbiota at week 17. (**C**) Distribution of differential abundant genus at week 17; the mean relative abundance is represented in log scale, all genera in the AOM/DSS group were different from those in the other two groups (all *p* < 0.05 by Mann–Whitney test, and *n* = 3 in vehicle and AOM/DSS groups and *n* = 4 for in AOM/DSS+M9 group).

**Figure 6 nutrients-13-01143-f006:**
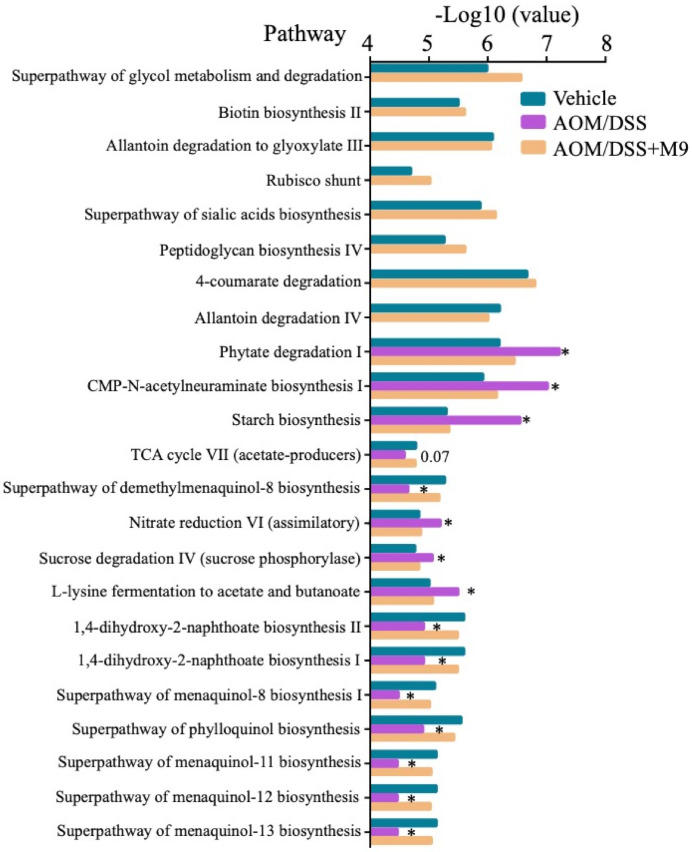
The difference in metabolic pathways among three groups on week 17. * *p* < 0.05 represents the difference between AOM/DSS and Probio-M9 groups. *n* = 3 in vehicle and AOM/DSS groups and *n* = 4 in AOM/DSS+M9 group; CMP: Cytosine nucleotides, TCA: the tricarboxylic acid cycle.

**Table 1 nutrients-13-01143-t001:** Inflammation score calculation criteria.

	Score	0	1	2	3	4
**I**	Inflammation	none	mild	moderate	severe	---
**E**	Extend	none	mucosa and submucosa	transmural	---	---
**R**	Regeneration	complete regeneration	almost complete regeneration	regeneration with crypt deletion	surface epithelium not intact	no tissue repair
**C**	Crypt damage	none	1/3 of basal damaged	2/3 of basal damaged	only surface epithelium intact	entire crypt and epithelium lost
**P**	Percent involvement	none	1–25%	26–50%	51–75%	76–100%

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
