# Peer review of "Inhibitory Effects of Breast Milk-Derived Lactobacillus rhamnosus Probio-M9 on Colitis-Associated Carcinogenesis by Restoration of the Gut Microbiota in a Mouse Model"

_nutrients, 2021, doi:10.3390/nu13041143_

Round 1

Reviewer 1 Report

Dear authors, 

I appreciate the data presented in your study, however I have some suggestion, comments and questions.

Why did you decide not to measure the colonic cytokines by ELISA or RT-PCR ? The cytokines, such as IL-1β, IL-6, IL-12, IL-17, IL-10, TNF-a and TFG-b have a role in the inflammatory process and probiotic strains are involved in their modulations.  Thus, you can correlate the changes in gene expression profiles with microbiota alterations by probiotic Probio-M9 administration. Please see:

Flemer B, Lynch DB, Brown JM, et al. Tumour‐associated and non‐tumour‐associated microbiota in colorectal cancerGut. 2017;66:633‐643

Bassaganya‐Riera J, Viladomiu M, Pedragosa M, De Simone C, Hontecillas R. Immunoregulatory mechanisms underlying prevention of colitis‐associated colorectal cancer by probiotic bacteriaPLoS ONE. 2012;7:e34676. 

The M&M section lacks references, especially in the histological analysis section. Please add references throughout the section.

In the AOM/DSS mice model, it is possible to find polyps in the mice intestine (colon). Did you record in the intestinal tissue the location, number and diameters of polyps in the colon?

I suggest adding a shared and unique OTU venn diagram among different groups.

In the line 376 - 378 : To confirm the hypothesis of colonization of the intestine by the strain Probio-M9 you must identify them in the feces of the animals. Please do this experiment to state about colonization.

Author Response

Thank you very much for the helpful comments on our manuscript. We believe that the novelty of the present study resides in demonstrating the anti-inflammation effects of Probio-M9 as well as their anti-tumor genesis effects in inflammatory bowel disease. In particular, demonstrating the microbiota alterations effects of Probio-M9 is considered to provide a new mechanistic insight into physiological function of breast milk.

Reviewer 2 Report

-did the authors detect differences in apoptosis or wnt /bcatenin signaling activation upon M9? Could the authors show the staining for cleaved caspase 3 and WB for bcatening P-GSK3b?

- in figure 2D, please add also westernblot for total AKT and STAT3/

- given the important role of microbiota in the regulation of T cell function and differentiation, it is important to show the effect of M9 on T cells in intestinal mucosa (especially CD8 cells in tumor regions)

- do the authors have the chance to perform plasma metabolomics given the changes observed in microbiota metabolic pathway? Could the author access the concentration of fecal SCFA?

Author Response

Thank you very much for the favorable comments on our manuscript. We appreciate your efforts and time in reviewing our manuscript. Please find a detailed point-by-point reply to each comment below. Please note that the changes we made according to reviewer comment are highlighted in yellow in the revised manuscript.

Round 2

Reviewer 1 Report

Dear Authors, 

I appreciate the effort to answer questions and the manuscript was corrected accordingly.

Sincerely